# Molecular and Genetic Advances in Small Cell Lung Cancer Landscape: From Homogeneity to Diversity

**DOI:** 10.3390/ijms25010224

**Published:** 2023-12-22

**Authors:** Lodovica Zullo, Filippo Gustavo Dall’Olio, Giovanni Rossi, Chiara Dellepiane, Giulia Barletta, Elisa Bennicelli, Marta Ingaliso, Marco Tagliamento, Carlo Genova

**Affiliations:** 1Dipartimento di Medicina Sperimentale (DIMES), Università Degli Studi di Genova, Via Leon Battista Alberti 2, 16132 Genova, Italy; lodovica.zullo@gustaveroussy.fr; 2Departement de Medicine Oncologique, Institut Gustave Roussy, 114 Rue Edouard Vaillant, 94800 Villejuif, France; filippogustavo.dall-olio@gustaveroussy.fr; 3Oncologia Medica 2, IRCCS Ospedale Policlinico San Martino, Largo Rosanna Benzi 10, 16132 Genova, Italy; giovanni.rossi@hsanmartino.it (G.R.); chiara.dellepiane@hsanmartino.it (C.D.); giulia.barletta@yahoo.it (G.B.); elisa.bennicelli@hsanmartino.it (E.B.); 4Dipartimento di Scienze Chirurgiche e Diagnostiche Integrate (DISC), Divisione di Anatomia Patologica, Università degli Studi di Genova, Largo Rosanna Benzi 10, 16132 Genova, Italy; marta.ingaliso@alice.it; 5Clinica di Oncologia Medica, IRCCS Ospedale Policlinico San Martino, Largo Rosanna Benzi 10, 16132 Genova, Italy; 6Dipartimento di Medicina Interna e Specialità Mediche, Università Degli Studi di Genova, Viale Benedetto XV 6, 16132 Genova, Italy

**Keywords:** small cell lung cancer, SCLC, never smokers, NGS, liquid biopsy, ctDNA

## Abstract

Small cell lung cancer (SCLC) has been historically considered a homogeneous disease and thus approached as a single entity when it comes to clinical studies design and new treatments developments. However, increasing knowledge in the genetic and molecular landscape of this disease challenges this concept, opening the possibility that different subtypes might show differential vulnerability to treatments. In this narrative review, we gather the most relevant advances in genetic and molecular characterization of SCLC, focusing on how these discoveries may be used to design the path for a personalized treatment approach. Indeed, we discuss the new classification based on differential protein expression, the prevalence and significance of oncogenic drivers (e.g., *EGFR* mutations and *ALK* rearrangements) in SCLC, the genetic characteristics of SCLC in patients with no smoking history, and the existing evidence supporting the use of liquid biopsy for capturing the heterogeneity of the disease. We use the keywords “small cell lung cancer”, “SCLC”, “EGFR”, “ALK”, “histological transformation”, and “transcriptional factors” to identify original research manuscripts, clinical trials, case reports, and case series from PubMed.

## 1. Introduction

Small cell lung cancer (SCLC) accounts for 10–15% of all lung cancer cases, with a prevalence of 1–5 cases per 10,000 people in Europe, and it is characterized by the strongest connection with smoking habit among all the lung cancer histologies [1].

Due to rapid growth and early metastatic spread, SCLC does not benefit from low-dose computed tomography (CT) screening [2,3]. Overall survival (OS) for the extended disease is extremely poor (<10% at five years) [1]. Unlike what has occurred with non-small cell lung cancer (NSCLC), the introduction of novel therapies into the treatment paradigm has been limited, and even when progress has been made, as with immunotherapy, the benefits achieved have generally been modest [1,4,5,6].

SCLC has traditionally been viewed as a homogeneous disease, shaping treatment strategies accordingly. The standard approach has involved platinum-based chemotherapy, resulting in rapid and profound responses but rarely achieving long-term durability [1]. Recent translational research has begun to challenge this paradigm, fueling increasing interest in the molecular subtypes of SCLC and their potential implications for therapeutic strategies. Indeed, differential gene expression in the different molecular subtypes and during the disease course might influence sensitivity and resistance to several therapeutic agents. The identification of predictors of responses to immunotherapy is of extreme clinical importance since the addition of an anti-programmed death-ligand 1 (anti PD-L1) to the platinum–etoposide backbone represents the new standard of care for frontline treatment of ED-SCLC, regardless of PD-L1 or any other biomarker [4,5]. The addition of atezolizumab and durvalumab to first-line chemotherapy treatment allows for a gain of two–three months in overall survival (OS), increasing from 10 to 12–13 months of survival. However, currently, there are no validated biomarkers that can identify those (few) patients who may achieve long-term benefits from frontline chemo-immunotherapy [4,5]. Furthermore, novel targets, such as delta-like ligand 3 (DLL3), are of great interest for the potential development of new classes of drugs (antibody–drug conjugates; bispecific proteins) that specifically recognize them [7,8].

Finally, a small but significant percentage of SCLC cases occur in patients with no history of smoking, often with recognized oncogenic drivers (e.g., *EGFR* sensitizing mutations) [9,10,11,12]. Due to the rarity of this occurrence, reports on disease course and treatment responses are mostly anecdotal and provide limited information.

This review summarizes relevant updates involving the transcriptomic and genomic characterization of SCLC, both in terms of biological findings and clinical implications, with the objective of defining the evolving molecular landscape of this disease. Within the context of this review, we included a specific focus on potentially actionable oncogenic drivers and novel targets for drug development.

### 1.1. Challenging the Concept of SCLC as a Homogenous Disease

SCLC has been traditionally considered a homogeneous disease, both from a pathological and clinical point of view, with a consistent tendency to high response rate to frontline chemotherapy but, nevertheless, an extremely poor prognosis, with rare exceptions.

On a molecular basis, SCLCs show almost universal inactivation of *tumor protein (TP53)* and *retinoblastoma protein (RB1)* genes [4,5]. However, the heterogeneity of SCLC exists and lies more in the differential expression of transcription factors than in gene-level alterations. This paragraph will primarily discuss the proposed classification based on differential gene expression and transcriptional factors, with reference to recent, comprehensive genomic studies on SCLC.

SCLC has been traditionally divided into “classic” and “variant” morphological subtypes [13], with morphological differences being driven by differential expression of *achaete-scute homolog 1* (*ASCL1*), also known as *ASH1*, and *neurogenic differentiation 1 (NeuroD1)*, two master transcription factors that govern neuroendocrine differentiation [14]. Gathering methylation-based and transcriptomic data from several studies, Rudin and colleagues proposed a classification focused on the differential expression of four key transcription regulators: *ASCL1*, *NeuroD1*, *yes-associated protein 1 (YAP1)*, and *POU class 2 homeobox 3 (POU2F3).* The four proposed categories are SCLC-A, SCLC-N, SCLC-Y, and SCLC-P, with the last letter referring to the transcription regulator whose expression is most strongly associated with each subtype [14].

The SCLC-A subtype is characterized by a high expression of *ASCL1* and a low expression of *NeuroD1*. It shows a neuroendocrine phenotype and is the most common subtype of SCLC, representing around 70% of all cases. SCLC-A tumors have high expression of *MYC* and *thyroid transcription factor-1* (*TTF-1*).

The SCLC-N subtype is characterized by a high expression of *NeuroD1* and a low expression of *ASCL1* and is a less common subtype than SCLC-A. SCLC-N tumors show a high expression of *DLL3*.

The SCLC-P subtype is characterized by a high expression of *POU2F3* and a low expression of *ASCL1* and *NeuroD1*. This subtype is associated with a non-neuroendocrine, squamous cell carcinoma-like phenotype, and it is a less common subtype of SCLC. SCLC-P tumors have a high expression of genes involved in squamous cell differentiation.

Finally, the fourth proposed subtype, SCLC-Y, is characterized by a high expression of the *YAP1* gene and a low expression of *ASCL1* and *NeuroD1*. Interestingly, in *YAP1*-expressing cell lines, *RB1* knockdown leads to a loss of *YAP1* expression, whereas *YAP1* knockdown has no effect on *RB1* expression, suggesting that *RB1* is upstream of *YAP1*. The SCLC-Y subtype is associated with a mesenchymal phenotype, and its frequency is the lowest [15].

Immunohistochemistry (IHC) analysis confirmed the classification based on *ASCL1, NeuroD1*, and *POU2F3* differential expression. However, the uniqueness of the fourth class, SCLC-Y, has been questioned, as *YAP1* seems to be expressed at low levels, primarily in combined SCLC. Furthermore, the resulting protein is broadly distributed across all subgroups, and although it is enriched in ASCL1-NEUROD1-negative cases, its expression is comparable between POU2F3-positive and POU2F3-negative subgroups [16]. In a study conducted on 146 tumor microarrays containing 194 cores from SCLCs (all primary tumors), Qu and colleagues showed that IHC was capable of distinguishing among subtypes based on ASCL1, NEUROD1, and POU2F3. However, a clear positivity for YAP1 was found in only 2.8% (*n* = 4) of the tumors, and only 3 out of 146 tumors (2%) were uniquely positive for YAP1. Moreover, 6.3% (*n* = 9) of the samples were negative for all the markers (NAPY-) [17].

Similar results are reported by Caeser et al. on 37 patient-derived xenografts (PDXs)/circulating tumor cell (CTC)-derived xenografts (CDXs) analyzed by IHC. Interestingly, even in the cases where RNA sequencing showed expression of *YAP1*, protein staining was not confirmed by IHC [18]. Furthermore, when they performed principal component analysis (PCA), they identified four clusters based on the presence of ASCL1, NEUROD1, and POU2F3 but not on the presence of YAP1. These clusters were ASCL1-driven, ASCL1/NEUROD1-driven, NEUROD1-driven, and POU2F3-driven, respectively [18].

Recently, another categorization aiming to identify patients who would benefit more from immune checkpoint inhibitors (ICIs) was validated in the IMpower 133 trial [4]. The SCLC-I subtype, where “I” stands for “inflamed”, replaced SCLC-Y [19]. This subgroup is characterized by higher immune infiltration and elevated expression of several genes encoding immune checkpoint molecules, including programmed-death 1 (PD-1), PD-L1, and Cytotoxic T-Lymphocyte Antigen 4 (CTLA4). These findings support the presence of an inflamed microenvironment primed for a response to immune checkpoint inhibitors (ICIs). Furthermore, the high expression of major histocompatibility complex class I (MHC I) appears to be associated with a durable response to ICIs. MHC I expression is typically epigenetically downregulated in most SCLC cases; however, a small yet unique subset with a non-neuroendocrine phenotype (ASCL1- and NEUROD1-negative) may de-repress MHC I expression, leading to significant clinical benefits from immunotherapy [20].

Albeit intriguing from a biological point of view, the investigation of the clinical implications of this subgrouping is still ongoing.

Efforts have been made to define and exploit the distinct therapeutic vulnerabilities within these SCLC subtypes to design targeted approaches that could ultimately enhance clinical outcomes for patients with this disease. For instance, *DLL3*, which is a promising target for SCLC treatment, with different compounds under development [21], is reportedly more expressed in the SCLC-A subtype [22], while an enhanced sensitivity to Aurora kinase inhibitors could be seen in the MYC-high NEUROD1-high SCLC subtype (SCLC-N) [23]. Aurora kinases regulate cell cycles by being implied in the G2-M transition. The rationale for the use of specific inhibitors of Aurora kinases lies in the fact that, as both TP53 and Rb play key roles in the regulation of cell cycle progression, most of the SCLCs show dysregulation in the cell cycle [23,24,25].

A recent large-scale real-world analysis of 3600 SCLC tissue biopsies, analyzed via Foundation Medicine Inc., investigated the genomic landscape of SCLC and explored the prognostic impact of different molecular alterations [26]. This study confirmed previously reported loss-of-function alterations in *KMT2D*/*MLL2* (12.9%), *CREBBP* (6.1%), and *NOTCH1* (5.9%), and gain-of-function events and copy number amplifications in *MYC* (6.0%), *MYCL* (7.2%), and *SOX2* (3.4%) as frequent molecular alterations in SCLC. Interestingly, the large scale of this study allowed us to detect gene alterations that had not previously been associated with SCLC. For instance, inactivating mutations in *KEAP1* were detected in ≈3% of tumor samples. Other tumor suppressors such as *TET2*, *SMARCA4*, and *STK11* were found to be inactivated in ≈2–3% of samples.

Based on the recurrent and mutually exclusive molecular findings, the authors proposed three novel genomic subtypes:Tumors without inactivation of *TP53/RB1*;Tumors harboring inactivating mutations in *STK11*;Tumors harboring oncogenic drivers that are usually associated with NSCLC.

*TP53*-inactivating mutations appear to be mutually exclusive with amplifications in *MDM2*, a negative regulator of p53, whereas *RB1* inactivation is mutually exclusive with inactivating alteration of *CDKN2A*, which positively regulates Rb, and activating mutations of *CCND1*, which negatively regulates Rb.

Moreover, *TP53* and *RB1* wild-type tumors have fewer smoke-associated genomic signatures and a higher percentage of alterations in *KEAP1*, *BRAF*, *KRAS*, and *FGFR1*.

In the absence of molecular alterations of *TP53* and *RB1*, other factors might induce protein inactivation. The authors explored the potential role of human papillomavirus (HPV) infection in the functional inactivation of p53 and Rb, specifically via the E7 oncoprotein, which inactivates RB family p107 and p130. In the HPV+ subpopulation, the rates of *TP53/RB1* alterations were lower than in HPV- patients: 70% vs. 92% for *TP53* and 56% vs. 74% for *RB1*.

The STK11-mutant subgroup shows a reduced percentage of RB1 losses and an enrichment of mutations in KRAS and KEAP1 [26].

SCLC harboring oncogenic drivers will be discussed in the following section.

### 1.2. Identification of Molecular Targets in Small Cell Lung Cancer

In recent years, our understanding of molecular profiling for SCLC has been steadily expanding, revealing potential genetic alterations that can be identified via various molecular biology techniques and panels despite the absence of approved targeted treatments for the disease. In a study including 59 patients with SCLC and 1 with combined SCLC histology (not otherwise specified), both in extensive- (*n* = 27) or limited-stage (*n* = 33), the assessment of molecular aberration revealed the presence of 1 *BRAF* V600E mutation in 1 patient, out of 46 samples analyzed for *BRAF* alterations. In the cases where the analysis for *EGFR* (*n* = 31), *KRAS* (*n* = 35), *NRAS* (*n* = 37) mutations, *ALK* rearrangement (*n* = 58), or *MET* amplification (*n* = 42) was possible, no alteration was detected. All patients in this series had a smoking history [27].

A study of similar size, conducted in a Japanese population, showed that 15% of patients (*n* = 9 out of 60) had at least one molecular aberration. Only two patients were never smokers, whereas the rest had a smoking history. Specifically, one *EGFR* mutation (G719A), one *KRAS* mutation (G12D), three *PIK3CA* mutations (E542K, E545K, and E545Q), one *AKT1* mutation (E17K), one *MET*, and six *PIK3CA* amplifications were detected. All these molecular alterations were found in patients with a smoking history. Both *EGFR* and *KRAS* mutations were detected in patients with a combined histology (SCLC plus adenocarcinoma). In this cohort, 31 patients had limited-stage and 29 had extensive-stage disease [28].

In another Asiatic cohort comprising 30 patients with SCLC (20 in extended- and 10 in limited-stage), 33% (*n* = 10) of tumors had molecular alterations, including six *EGFR*, two *PTEN*, one *KRAS*, and one *PIK3CA* mutations. Among these patients, nine had a smoking history, whereas only one, whose tumor harbored an *EGFR* mutation, had no smoking history. No molecular alteration, among those searched for, was detected in the remaining three patients with no smoking history [29].

Cardona et al. performed multigene profiling via next-generation sequencing (NGS) in 10 never/ever and 10 smoker patients with SCLC and matched by clinical and pathological features. Three *EGFR* sensitizing mutations were found, all in never/ever smoking patients (never/ever smoking vs. smoking patients *p* = 0.01). Tumor mutational burden (TMB) did not statistically differ between the two subgroups (*p* = 0.18), but the most common molecular alterations found in the two cohorts were different: for instance, higher prevalence of *EGFR* mutations (30% vs. 0%) was found in never/ever smokers, whereas a greater frequency of *RB1* mutations (80% vs. 40%) was detected in patients with smoking history [9].

In a series of 11 cases of patients with no or little smoking history (≤10 packs/year), whose tumors were analyzed via targeted NGS, mutations of *EGFR*, *BRCA1*, *KRAS*, *NRAS*, and *ATM*, amplification of *MET*, and *TMPRSS2-ERG* fusion were detected. However, a subsequent histopathological second look revealed combined histology (the two cases harboring *EGFR* mutations), intermediate histology between large cell neuroendocrine carcinoma and SCLC (the cases harboring *ATM* mutation and *MET* amplification), a grade 2 neuroendocrine tumor/atypical carcinoid of either pancreatic or pulmonary origin (the case harboring *BRCA1* mutation), a suspected hepatobiliary origin (the case harboring *KRAS* mutation), and a suspected prostatic origin (the case harboring *TMPRSS2-ERG* fusion). The case with an *NRAS* mutation was not confirmed to be an SCLC. Only three cases were confirmed as SCLC, and two of them harbored *TP53* and *RB1* mutations [10].

In another series of 27 Asian patients with no smoking history, NGS analysis revealed three *EGFR* mutations (one exon 19 deletion, G719A, and L858R), and another *EGFR L858R* mutation was identified via polymerase chain reaction (PCR). Other molecular alterations detected included *TP53* (*n* = 26), *RB1* (*n* = 7), *PTEN* (*n* = 5), *MET* (*n* = 4), and *SMAD4* (*n* = 3) mutations [12].

As previously mentioned, a recent large-scale NGS analysis was performed on 3600 real-world SCLC cases [26]. Out of the 678 patients for whom clinical data were available, 60 (9%) had a tumor genomic profiling via Foundation Medicine before receiving any treatment, thus depicting the molecular landscape at baseline. In the whole cohort, compared to previous studies, genes of the PI3K and RAS-RAF-MAPK pathways were more likely altered (*PTEN* 9.9%, *PIK3CA* 5.6%, *RICTOR* 5.6%, *EGFR* 3.4%, *KRAS* 3.3%, and *NF1* 3.3%). Interestingly, the authors proposed a new genomic subgroup of SCLCs based on the presence of oncogenic drivers. Indeed, beyond the already mentioned mutations of *EGFR, ALK* (*n* = 5), *RET* (*n* = 5), *ROS1* (*n* = 3), and *NTRK1* (*n* = 1) oncogenic molecular alterations were detected. The authors suggest that both the *EGFR* mutant and those harboring other oncogenic drivers derive from a sub-clonal evolution of oncogene-addicted NSCLCs, keeping the original driver but acquiring further molecular alterations that are typical of de novo SCLC, as mutations in *TP53* and *RB1* [26].

Details regarding the tested molecular alterations, the techniques used, and the molecular findings of the reported studies are reported in Table 1.

### 1.3. Targeting EGFR Mutations and ALK Rearrangements in Small Cell Lung Cancer

The limited experiences of targeted therapies for the treatment of SCLC regard almost exclusively the use of EGFR and ALK inhibitors (EGFRis and ALKis) in the presence of specific molecular alterations of the corresponding genes.

*EGFR* mutations and *ALK* rearrangements have been described in cases of histological transformation as a mechanism of resistance to tyrosine kinase inhibitors (TKIs) [11,32,33,34], but de novo alterations of these genes in SCLC are unusual.

*EGFR* mutations in SCLC are reported, ranging from 2.6% to a maximum of 7% in a Chinese dataset [35,36,37].

In NSCLC, *EGFR* alterations are more common in Asian ethnicity and female patients with no smoking history [35,38], whereas their relative distribution in patients affected by SCLC is less clear. Some studies report no correlation between *EGFR* mutation prevalence and clinical features, such as gender, age, or clinical stage at diagnosis [35,36]. As regards tobacco, they appear to be more frequent in patients with no smoking history [9,37].

Araki and colleagues reported the case of a female patient with no smoking history and a performance status (PS) ECOG of 3 at diagnosis, who received gefitinib 250 mg/day as a first-line treatment for an SCLC harboring *EGFR* delE746-A750 of exon 19, with rapid improvement in symptoms and partial response (PR) in the first radiological evaluation. Progression-free survival (PFS) reached almost five months when the patient suddenly died of cerebral hemorrhage [39,40].

Shiao et al. reported two cases of SCLC harboring, respectively, *EGFR* delE746-S752insV and *EGFR* delE746-A750, two deletions in exon 19. The first one, a male patient with a smoking habit, never received a TKI and died 10 months after the diagnosis; the second one, a 54-year-old female patient without a smoking history, received gefitinib in subsequent line, with PD as the best response [37].

Gefitinib was also used for the treatment of a female patient with no smoking history who developed a SCLC with an *EGFR* exon 19 deletion, together with *RET* E616K and multiple *TP53* mutations. No tumor shrinkage was observed, and no survival outcomes were reported [12].

Le et colleagues reported the case of a female patient with no smoking history, affected by SCLC harboring *EGFR* delL747_P753insS, who received erlotinib 150 mg/day after progression to standard platinum-based chemotherapy. Unfortunately, the patient experienced rapid progression and died within two months from the start of erlotinib. Interestingly, when IHC for EGFR was performed, no expression of the protein was detected on tumor cells [41].

Another case of *EGFR*-mutant SCLC resistant to erlotinib was described by Petricevic et al. This was the case of an 84-year-old Caucasian woman with no smoking history. NGS analysis detected an *EGFR* exon 19 deletion in both tissue and blood samples, plus *TP53* and *PIK3CA* mutations. At diagnosis, PS ECOG was 3 due to cancer-related symptoms. The patient started a first line of carboplatin area under the curve (AUC) 6 every three weeks, with PD at the first radiological evaluation. Then, a second line with dose-reduced erlotinib (100 mg/day) was started. However, rapid progression occurred after eight weeks, and the treatment was discontinued [42].

Batra et al. reported the case of a young male patient with no smoking history with a composite tumor (adenocarcinoma and SCLC) harboring an *EGFR* exon 19 deletion. After an underwhelming response to frontline chemotherapy, the patient underwent osimertinib, with a rapid PR and good tolerance. Six months after the treatment started, the patient was still progression-free and under treatment [43]. In a similar case of composite lung cancer, with both the adenocarcinoma and SCLC components harboring *EGFR* exon 19 deletion, *TP53,* and *RB1* loss, osimertinib obtained a PFS of eight months and, continued beyond progression, pleural effusion control for over 12 months [44].

Evidence on SCLC harboring *ALK* rearrangements is even more limited.

In a series of 142 consecutive cases of treatment-naïve SCLC, 11% (*n* = 16) showed expression of ALK in IHC. However, the expression was focal and of lower intensity (3+ intensity was not detected in any case) compared to the diffuse and of higher intensity expression generally showed in NSCLC. Moreover, at subsequential analyses via fluorescence in situ hybridization (FISH) and PCR performed in ALK-expressed SCLC, no activating alterations (rearrangements, point mutations, or amplification) were detected, and in the cases where copy number gains were shown (4/12), they were mild, accounting for 3–5 copy increases. The absence of activating mutations behind the expression of the protein suggests that only a normal form of ALK is expressed. Thus, ALK expression should not be considered as a surrogate of the presence of a molecular target in SCLC [45].

As regards the possibility of using ALKi as targeted therapy in selected cases of SCLC, a clinical case reported rapid PR to alectinib, even though in combination with irinotecan, as second line for a 26-year-old patient with SCLC. Indeed, disease progression after only two cycles of platinum-based chemotherapy and the young age of the patient led to NGS analysis, which detected an *ALK* rearrangement. The combination of alectinib (600 mg BID) and irinotecan was administered, rapidly achieving both radiological and molecular response as assessed per NGS on circulating tumor DNA—ctDNA (variant allele frequency-VAF reduced from basal 19% to 0.6% after two cycles). PFS reached 6 months [46].

A retrospective study included 31 patients with a combined SCLC-adenocarcinoma, harboring driver alterations. The median age was 58 (range 29–73) years, 48% (*n* = 15) were female, and 55% (*n* = 17) had no smoking history. The most represented molecular alterations were *EGFR* sensitizing mutations (74%, *n* = 23), most commonly exon 21 L858R (39%, *n* = 12), followed by *ALK* fusions (9.7%, *n* = 3), *BRAF* mutations (6.5%, *n* = 2), *ROS1* fusions (6.5%, *n* = 2), and *RET* fusions (3.2%, *n* = 1). All patients with *ALK* and *RET* fusions and *BRAF* mutations were treated with first-line chemotherapy; one patient with *ROS1* fusion was treated with first-line targeted monotherapy; six patients with *EGFR* mutations (exon 19 deletion and L858R) were treated with first-line targeted therapy, whereas eight patients (four with exon 19 deletion, three with L858R, and one with exon 19 G719X) were treated with first-line targeted therapy plus chemotherapy. ORR was 43% (*n* = 3) for patients treated with first-line targeted therapy, 43.8% (*n* = 7) for patients treated with first-line chemotherapy, and 62.5% (*n* = 5) for patients treated with first-line combination therapy. The median PFS were 5.0, 4.0, and 7.9 months, respectively (*p* = 0.02); the median OS were 17.4, 14.1, and 12.9 months, respectively *(p* = 0.3). Granular data regarding response and survival outcomes to specific inhibitors used for specific molecular alterations are not available [47].

Data regarding the use of targeted agents for the treatment of SCLC harboring oncogenic drivers are gathered in Table 2.

### 1.4. Small Cell Lung Cancer in Patients with No Smoking History

SCLC histology is known to be strongly linked to smoking habits [48]. However, 2–3% of SCLC occurs in patients with no smoking history [31,49], and an even higher prevalence (13–23%) among patients with no smoking habit is described in some Asian cohorts [12,50,51].

Among risk factors other than smoking, second-hand tobacco [49] and indoor radon exposure may play a role in the development of SCLC [52], even though the causal effect of radon exposure still needs to be determined.

In recent years, some studies have tried to assess whether clinical and genetic characteristics differ in SCLCs that develop in patients with or without tobacco habit. The hypothesis is that, following the paradigm of NSCLC, patients with no smoking history could develop a biologically different disease and thus benefit from a different management in terms of systemic treatments.

Globally, patients who never smoked and yet suffer from SCLC are often female [9,30,50,51,52,53] and usually older than patients with a smoking habit [9,50,51]; in the cohort reported by Tseng and colleagues, a higher percentage of patients without a smoking history compared to patients with a smoking history was diagnosed at ≥70 years old (57.3 vs. 44.8, *p* < 0.001 (χ2 test)) [51]. Moreover, patients without a tobacco habit tend to present more frequently with extensive disease [30,51].

The role of smoking history as a prognostic factor for SCLC is still not well defined, with some authors reporting an association between smoking habit and worse survival outcomes [9,12,53]. In conflict with these data, shorter survival is described for patients without a tobacco habit in other cohorts [50,51], whereas Thomas et al. reported no survival differences between patients with or without smoking history [30].

As previously mentioned, inactivating mutations of *TP53* and *RB1* are almost ubiquitarians in SCLC [54]. The disease is also characterized by a high mutational rate and genomic signatures of tobacco exposure [55,56]. However, recent studies suggest a different distribution of genetic alterations in tumors occurring in patients with or without smoking history and, interestingly, a higher prevalence of *EGFR* mutations in patients who never smoked. Indeed, Thomas et al. reported the results of targeted sequencing on nine tumors that developed in the absence of smoking history, showing that they were less likely to harbor *TP53* (*p* < 0.001) or *RB1* (*p* < 0.002) alterations compared to tumors of patients with tobacco habit (*n* = 320). On the contrary, tumors in patients who never smoked had a higher probability of harboring *EGFR* alterations (11% vs. 0.94%), although the difference was not statistically significant (*p* = 0.105). Moreover, TMB appears to be lower in patients without than in patients with smoking habits (*p* = 0.005) [30].

The authors also performed whole exome sequencing (WES) in four patients with no smoking history, revealing the presence of two sensitizing *EGFR* mutations (exon 19 deletions) and no evidence of tobacco-related signature (C > A). Compared to the tumors positive for signature-4 described by George et al. [54], these tumors had lower TMB (3.7 vs. 8.1 mut/Mb) and a lower rate of *TP53* mutation (50% vs. 89%) [30].

Cardona et al. confirmed the higher prevalence of *EGFR* mutations in the never/ever smoker population compared to their positive for smoking history counterparts (3 vs. 0, *p* = 0.01). TMB was not statistically different in the two groups (*p* = 0.18), each one comprising 10 patients [9].

In another cohort of 19 patients with no smoking habit and de novo SCLC, eight were tested for *EGFR* deletion of exon 19 and L858R in exon 21 via PCR. The assessment of *EGFR* mutations revealed one deletion in exon 19 and one L858R. In the cases tested for *KRAS* mutations (*n* = 8) or *ALK* rearrangement (*n* = 5), no alterations were found, whereas six out of seven patients tested for *RB* expression had *RB* loss [31].

In the cohort of 27 Asian patients with no smoking history described by Sun and colleagues, four *EGFR* mutations were detected, three via NGS and one via PCR [12] (Table 1). Moreover, *RET*, *VHL*, and *FBXW7* mutations were detected in patients with no tobacco habit, whereas no alteration in these genes was found in tumors of patients with a smoking history [12].

### 1.5. Liquid Biopsy: A Novel Tool for Deeper Knowledge?

An obstacle to the understanding of SCLC biology is the usual scarce tissue availability for molecular studies since SCLC rarely undergoes surgery [57,58]. Depicting the molecular landscape of the disease at resistance to platinum-based chemotherapy is even more challenging. Indeed, due to the rapid clinical worsening and the necessity of starting a new line of treatment, biopsies at progression are rarely performed in clinical practice, thus hampering the acquisition of knowledge regarding mechanisms of resistance to treatments [59,60,61]. In this context, liquid biopsy could be a useful tool by capturing the spatial and temporal molecular heterogeneity of the disease.

Research in the field of liquid biopsy in SCLC has started with the detection and characterization of CTCs, which have been demonstrated to be prognostic biomarkers [62], and it is moving forward to the possibility of deducing the complexity of molecular landscape from ctDNA [63]. Alterations in *TP53* and *RB1* are also the most frequently detected in blood samples [58,60,63], consistent with results obtained from tissue analysis [64,65]. Molecular aberrations involved in DNA damage repair (DDR) [60,63], chromatin and transcription regulation [63,66], NOTCH signaling [61,63,66], PI3K/AKT/mTOR [58,60,66], RTK/RAS/RAF [60] pathways, and copy number alteration of *MYC* [60,61,63,66] and *SOX2* [63] were also frequently detected in different reports.

The largest cohort of NGS testing on ctDNA obtained from SCLC patients was described by Devarakonda et al. Out of 609 liquid biopsies obtained from 564 patients, at least one non-synonymous mutation or amplification was detected in 91% of the samples, with mutations of *TP53* (72%) and alterations in *RB1* (18%) being the most common. Interestingly, *EGFR* alterations (mostly missense mutations) were found in 6% (*n* = 34) of the samples. Comparing baseline samples to those obtained at relapse, a higher frequency of alterations in the *androgen receptor AR)* gene that are potentially targetable was observed. Another potentially targetable pathway involved in relapse is the DDR one [60].

A smaller cohort of 43 samples, obtained from 22 patients and tested with deep sequencing of 430 genes, confirmed *TP53* and *RB1* as the most frequent mutated genes, respectively, altered in 91% and 64% of patients. Other commonly altered genes were *NOTCH1-4*, *EP300*, and *CREBBP*; copy number alterations of *MYC*, *MYCN*, and *MYCL1* were observed. For eight patients, tissue samples were available for matched molecular profiling using the same panel and the same platform. The comparison revealed that 94% of mutations found in tumor samples were also detected in ctDNA, suggesting that liquid biopsy could reliably reflect the molecular landscape of the tumor. Moreover, some somatic mutations were detected exclusively in ctDNA, and when subclonal architecture was inferred from VAFs, more mutation clusters were detected in blood samples rather than on tissue. In the same cohort, blood samples collected at different timepoints were available for 11 patients, depicting a change in time of the VAF of several mutation clusters before and after treatment. Interestingly, some subclones, initially “cleared” from chemotherapy, remerged at the time of progression, and among 33 mutations detected only at progression, nine are related to chemoresistance in solid and hematologic tumors [61].

Dynamic changes in SCLC-derived cell-free DNA (cfDNA) obtained before, during, and after treatment were also described by Almodovar et al. for 25 patients. These changes were correlated to response and relapse to therapy. In two cases, the reappearance of mutations in *TP53* and *NOTCH3*, which was first detected at baseline and then disappeared in concomitance to response to treatment, occurred before radiological progression was evident [58].

In the cohort described by Mohan et al., 6 out of 62 (9%) patients were longitudinally followed via liquid biopsy. In two cases, disease detection via copy number alteration or targeted sequencing permitted the recognition of disease progression before it was radiologically evident [63]. Taken together, these data suggest that liquid biopsy, especially the targeted sequencing of *TP53*, which is the most common mutated gene, could be a useful companion to radiological surveillance for patients suffering from SCLC. In addition to depicting dynamic changes in time, liquid biopsy may also capture the tumoral spatial heterogeneity, with several molecular aberrations that reflect a complex tumoral biology [61,63,66].

Another potential application of liquid biopsy is the possibility of predicting sensitivity or refractoriness to chemotherapy via the differential expression of molecular alterations directly on cfDNA or on CTC-derived models. While *Adenomatous polyposis coli (APC)* mutations seem to occur more frequently in baseline samples obtained from chemosensitive rather than chemorefractory SCLC (69% vs. 17%), baseline mutations in *TP53* (67% vs. 6%), *ataxiateleangectasya mutated (ATM)* (67% vs. 13%), and *folliculin (FANC)* (42% vs. 0%) are more frequent among samples obtained from chemorefractory SCLCs. In this study conducted on cfDNA, refractoriness was defined as progression occurring within 90 days from the end of the first-line regimen. This study was conducted only on small cohorts of 16 chemosensitive and 12 chemorefractory patients [67,68].

CDXs might effectively mirror the spatial and temporal heterogeneity of SCLC, which can be fully elucidated via the analysis of differential single-cell expression patterns [69]. In a study involving 14 samples, single-cell transcriptomics revealed the presence of multiple resistant clusters with distinct expression profiles at the onset of platinum resistance. In contrast, samples obtained from baseline CTCs exhibited greater homogeneity. These findings suggest a global need to optimize frontline treatments to maximize their effectiveness and duration before the emergence of heterogeneity renders the disease non-targetable via the inhibition of a single pathway. Indeed, targeting multiple pathways may be necessary to overcome platinum resistance, and potential therapeutic targets may include *PARP1*, *CHEK1*, Aurora kinase genes (*AURKA* and *AURKB*), and EMT-associated genes [69]. The expression of *DLL3* is also temporally heterogeneous, with a clear decrease in the cases of mesenchymal transition [69]. Agents targeting DLL3, including antibody–drug conjugates [8] and bi-specific T cell engagers [7], are under investigation in solid tumors, including SCLC.

## 2. Discussion

SCLC has traditionally been considered a homogeneous disease in terms of therapeutic options, and most clinical research approaches have been borrowed from the management of non-oncogene addicted non-small cell lung cancer (NSCLC); indeed, the current therapy of choice for extensive SCLC is represented by chemotherapy plus an anti-PD-L1 inhibitor. However, while the combination of chemotherapy and immunotherapy has substantially doubled the survival of NSCLC patients, the benefit in SCLC, albeit consistently demonstrated, has been remarkably more limited [4,5].

Identifying biomarkers that could predict durable response to immunotherapy in SCLC is of extremely clinical relevance. Among all subtypes, SCLC-I seems to have the richest microenvironment, with a higher absolute number of T-cells, Natural Killer (NK) cells, and macrophages, compared to the others [19]. Moreover, these tumors have a higher expression of Human Leucocyte Antigens (HLA) [54], of other genes linked to antigen presentation [54], of immune checkpoint molecules [19], and of 8-gene interferon-γ-related T cell gene expression profile (GEP) [19], which is an independent predictor of response to immunotherapy in solid tumors [70]. The identification of different SCLC subtypes (-A, -N, -P, and -I) based on gene expression profiles represents a relevant first step toward treatment personalization, as it might explain why the global benefit of chemo-immunotherapy is limited, but nonetheless, there is a proportion of patients experiencing prolonged benefit.

Preclinical studies suggest that various subtypes of SCLC may exhibit distinct vulnerabilities to platinum-based chemotherapy and other agents, offering opportunities for personalized treatment in SCLC. Gay et al. report that SCLC-P cell lines are sensitive to cisplatin, to PARP-inhibitors (PARPis), and to antimetabolites (anti-folates), whereas SCLC-N and especially SCLC-I cell lines are refractory to cisplatin. SCLC-A cell lines show a wide spectrum of sensitivity to platinum: high expression of *SLFN11* in SCLC-A cell lines is correlated with higher sensitivity to both cisplatin and PARPis, whereas low expression of *SLFN11* within this subgroup is accompanied by lower responsiveness to both platinum and PARP inhibition [19]. Indeed, *SLFN11* is a known predictor of response to platinum and PARPis in SCLC [71,72,73]. However, Gay et al. report that, in their study, vulnerability to platinum and PARPis shown by SCLC-P cell lines is not linked to expression levels of *SLFN11*; indeed, despite high sensitivity to these agents, expression of *SLFN11* in these cell lines is modest [19].

Different combinations of PARPis and chemotherapeutic agents have been tested in phase I/II studies, showing signals of anti-tumor activity [73,74,75,76]. The rationale for their combination lies in the fact that the chemotherapeutic agents might enhance the efficacy of PARP inhibition by causing single-strand DNA damage [74]. While veliparib plus platinum-based chemotherapy has been tested as a frontline treatment of SCLC [75,76], PARPi-chemotherapy combinations might find their place in the relapse setting when patients who initially responded to DNA-damaging agents eventually progress. In order to identify subsets of patients that might benefit the most from these combination strategies, as previously mentioned, the expression of *SLFN11* might serve as a predictive marker for response to both platinum-based chemotherapy and PARP inhibition, and the identification of common features associated with platinum-based resistance, and suggesting consequential PARPi chemotherapy resistance, is a huge area of research [73,74].

To note, olaparib plus durvalumab (anti-PD-L1) showed a dismal overall response rate (ORR) of 10% (*n* = 2 patients) in 19 evaluable patients with relapsed SCLC treated in a phase II trial [77], suggesting that the combination of an agent directly inducing DNA damage with a PARP inhibitor might be more effective for SCLC treatment.

Moreover, the high expression of DLL3 in the SCLC-A subtype [22] might eventually make it a promising target for agents such as tarlatamab, a bi-specific T cell engager, which acts by binding both DLL3 and CD3 [21]. Tarlatamab has shown promising antitumor activity in a phase II trial, including pre-treated SCLC, regardless of DLL3 expression [78].

It must be noted that while transcriptional factor-based categorization defines different subtypes with potentially different vulnerabilities to treatment, in some cases, tumors may not be assigned to a category, but they appear to be “mixed”. The hypothesis is that rather than being separated entities, these subtypes testify to the temporal evolution of SCLC, with differential expression of transcription factors leading to the prevalence of one phenotype or another in a certain moment [19].

Liquid biopsy could be useful for capturing spatial and temporal heterogeneity of SCLC. This tool has the potential to revolutionize the management of multiple solid tumors, and this is especially valid for SCLC since this malignancy is often diagnosed in a metastatic stage, hence potentially releasing circulating DNA within the bloodstream. Currently, the use of liquid biopsy in common clinical practice for SCLC is still limited since the registered treatments for this malignancy do not include targeted agents. However, as the clinical landscape of SCLC evolves, liquid biopsy is expected to become a routine tool in the next few years. Indeed, liquid biopsy will represent a robust source of circulating tumor-related biomarkers for personalized treatments. Additionally, the dynamic evaluation of the circulating biomarkers might represent a useful tool for response assessment during treatment. To date, studies incorporating liquid biopsy in SCLC management collected only small sample sizes, and only a few cases have been followed longitudinally with samples collected at different timepoints [61,63]. Thus, while the potentialities of liquid biopsy for early identification of resistance to frontline treatment and for capturing tumor heterogeneity at relapse are intriguing, they should be further investigated on a larger scale.

Another subject of discussion is represented by SCLC harboring oncogenic drivers with an acknowledged role from NSCLC studies, such as *EGFR* or *ALK*. The identification of such biomarkers immediately seemed promising, with the promise of borrowing targeted agents from NSCLC. However, direct translation from NSCLC to SCLC finds several obstacles:In the first place, SCLC is strongly linked with smoking habits, which, in turn, is generally associated with the absence of the most common oncogenic drivers.The proportion of patients harboring such genic alterations is low, even among the few patients with no smoking history and an SCLC.Most available data on targeted agents in this setting are anecdotal, based on case reports, and involve pre-treated patients or even unfit individuals. In any case, results with targeted therapy are generally unsatisfying.The rare cases of SCLC harboring an oncogenic driver (e.g., *EGFR* activating mutations) may derive from mixed histology [47] and/or histological transformation [25]. This may partially explain the underwhelming responses to TKIs; indeed, while the transformed SCLC keeps the original molecular alteration, protein expression may be impaired (39), thus reducing the sensitivity to specific inhibitors [25].

*EGFR*-mutant NSCLC transforming to neuroendocrine tumors after TKI treatment are usually treated with platinum etoposide, a regiment that yields a clinical response rate of 54% in a retrospective series [79]. In this retrospective cohort, EGFR-TKIs were administered in 52% of cases, but mostly either in concomitance or in maintenance after chemotherapy. Moreover, retrospective data suggest that transformed *EGFR*-mutant SCLC might be significantly sensitive to taxane-based regimens, even when administered in later lines [79]. Translating this evidence to de novo (or spontaneously transformed) SCLC, this entity should be treated with first-line platinum–etoposide and a specific inhibitor as maintenance treatment; at progression, a taxane-based strategy should be considered.

## 3. Conclusions

The molecular characterization of SCLC is a promising field, with hopefully the potential to improve the management of this malignancy. Known oncogenic drivers with a consolidated role in NSCLC seem to have a limited impact in SCLC due to low frequency and clinical evidence. However, the identification of multiple phenotypes based on differential protein expression provides the basis for further research for personalized treatment development.

## 4. Future Directions

As the knowledge of SCLC subtypes evolves and novel molecular targets are discovered, the analysis of the molecular landscape for this disease is expected to lead to the development of new therapeutic strategies, including investigational targeted agents or novel drug classes, such as bispecific T-cell engagers. In this context, liquid biopsy will be a useful tool to guide the treatment, both in terms of baseline therapeutic decision and longitudinal monitoring of disease response.

## Figures and Tables

**Table 1 ijms-25-00224-t001:** Potential therapeutic targets found in molecular studies conducted in SCLC.

Author	Sample Size	Gene Tested/Technique/Panel	Evaluable Sample N	Molecular Alterations N (%)
Sivakumar et al. [26]	3600	FoundationOne/FoundationOneCDx	1515 tested with FoundationOne 2085 tested with FoundationOneCDX	356 (9.9) *PTEN* alt 213 (6) *NOTCH1* alt 201 (5.6) *PIK3CA* alt 154 (4.3) *CCNE1* alt 149 (4.1) *CDKN2A* alt 141 (3.9) *FGFR1* alt 126 (3.5) *ARID1A* alt 124 (3.4) *EGFR* alt 118 (3.3) *KRAS* alt 118 (3.3) *NF1* alt 99 (2.7) *KIT* alt 97 (2.7) *NOTCH3* alt 82 (2.3) *PDGFRA* alt 63 (1.7) *ATM* alt 54 (1.5) *SMARCA4* alt 53 (1.5) *NOTCH2* alt 52 (1.4) *BRCA2* alt 49 (1.4) *ATR* alt 41 (1.1) *CHEK2* alt 37 (1) *AKT1* alt 36 (1) *BRCA1* alt *
Thomas et al. [30]	320 patients with a smoking history 9 patients with no smoking history, who had all pure SCLC	NGS (FoundationOne^®^CDx)	329	**No smoking history** 1 (11) *EGFR* rear 1 (11) TMB ≥ 20 muts/Mb 1 (11) TMB 5.5–19 muts/Mb 6 (66) TMB < 5.5 muts/Mb **Smoking history** 3 (1) *EGFR* alt (all combined with *TP53* alt) 16 (5) TMB ≥ 20 muts/Mb 209 (65) TMB 5.5–19 muts/Mb 85 TMB (26) < 5.5 muts/Mb
Abdelraouf et al. [27]	59 pure SCLC 1 combined	*EGFR*, *KRAS* (Real-Time PCR) *BRAF*, *NRAS* (Multiplex PCR) *ALK* rearrangements (FISH) *MET* amplification (FISH)	31 (for *EGFR*) 35 (for *KRAS*) 46 (for *BRAF*) 37 (for *NRAS*) 58 (for *ALK*) 42 (for *MET*)	1 (2.2) *BRAF* V600E
Wakuda et al. [28]	57 pure SCLC 3 combined	*EGFR*, *KRAS*, *BRAF*, *PIK3CA*, *NRAS*, *MEK1*, *AKT1*, *PTEN*, *HER2* muts (pyrosequencing plus capillary electrophoresis) *EGFR*, *MET*, *PIK3CA*, *FGFR1*, and *FGFR2* ampl (Real-Time PCR) *EML4-ALK*, *KIF5B-RET*, *CD74-ROS1*, *SLC34A2-ROS1* fusion genes (multiplex RT-PCR)	60	1 (1.7) *EGFR* G719A 1 (1.7) *KRAS* G12D 3 (5) *PIK3CA* (E542K, E545K, E545Q) 6 (10) *PIK3CA* ampl 1 (1.7) *AKT1* E17K 1 (1.7) *MET* ampl
Wang et al. [29]	30 SCLC	*EGFR* exon 18, exon 19, exon 20, exon 21 *KRAS* exon 2 *BRAF* exon15 *PTEN* exon 5, exon 6, exon 8, *PIK3CA* exon 9, exon 20 (PCR)	30	5 (16.7) *EGFR* exon 19 (3 p.E746_A750del; 1 p.L747_S752del; 1 p.K737_I740delinsPHWD) 1 (3.3) *EGFR* exon 21 (p.Q849_H850delinsHN) 1 (3.3) *KRAS* G12D 1 (3.3) *PTEN* F154fs*5
Sun et al. [12]	30 patients with no smoking history, SCLC	NGS (Ion AmpliSeq™ Cancer Hotspot Panel v2) Confirmation via PNAClamp™ *EGFR* Mutation Detection kit for never-smokers	27 (for AmpliSeq™) 28 (for PNAClamp™)	4 (14.8) total; 3 *EGFR* (1 del19, 1 L858R, 1 G719A) detected by both methods; 1 L858R in a case tested only by PNAClamp™) 4 (14.8) *MET* alt 3 (11.1) *PTEN* alt 2 (7.4) *KIT* alt 1 (3.7) *CKDN2A* alt
Caeser et al. [18]	18 Paired PDX/clinical samples of SCLC	NGS (MSK-IMPACT targeted tumor sequencing)	18	4 (22) *EGFR* miss mut 2 (11) *PIK3CA* miss mut 2 (11) *KRAS* miss mut 1 (5.5) *KRAS* ampl 2 (11) *PTEN* miss mut 1 (5.5) *PTEN* deep del 2 (11) *CDKN2A* miss mut 2 (11) *ERCC2* trunc mut 2 (11) *NF1* trunc mut
Cardona et al. [9]	10 with no smoking history 10 with smoking, SCLC	NGS (TruSight TumorTM 170) Additional EGFR Real-Time PCR (Cobas^®^ v2 probe) in never/ever smoker	20	**No smoking history** 2 (20) *EGFR* (del19, L858R) 5 (50) TMB 8–14 muts/Mb 4 (40) TMB > 15 muts/Mb **Smoking history** 0 (0) *EGFR* 5 (50) TMB 8–14 muts/Mb 1 (10) TMB > 15 muts/Mb
Varghese et al. [31]	19 SCLC	*EGFR* del19/L858R (PCR) *KRAS* exon 12–13 mass spectrometry (Sequenom Inc., San Diego, CA, USA) and direct sequencing *ALK* rear FISH (dual-color break-apart ALK probe, Abbott Molecular) or IHC (ALK-01 Ventana) *RB* loss IHC (clone 1F8, Leica Biosystems)	8 (for EGFR) 8 (for KRAS) 5 (for ALK) 7 (for RB1)	1(12) *EGFR* L858R (SCLC + ADK) 1(12) *EGFR* del19/*PIK3CA* E545K (pure SCLC) 6(86) *RB1* loss
Thomas et al. [30]	4 patients with no smoking history	WES	4	2 (50) *EGFR* del19 Median TMB 3,73 muts/Mb

Abbreviations: ADK: adenocarcinoma; Alt: alterations; amp: amplification; FISH: fluorescence in situ hybridization; Mb: megabase; miss: missense; mut: mutation; N: number; NGS: next-generation sequencing; PCR: polymerase chain reaction; RT-PCR: reverse transcription polymerase chain reaction; SCLC: rear: rearrangement; small-cell lung cancer; trunc: truncating, WES: whole exome sequencing. * Only molecular alterations with a prevalence ≥ 1% are reported.

**Table 2 ijms-25-00224-t002:** Main results obtained with targeted treatments in clinical cases of SCLC harboring oncogenic drivers.

Author	Targeted Molecular Alteration	Concomitant Molecular Alterations	Drug	Line of Treatment	Outcomes Reported
Takuma et al. [44] *	*EGFR* exon 19 del	*TP53* loss, *RB1* loss	osimertinib	1	BR: PR PFS: 8 months
Batra et al. [43] *	*EGFR* delE746_A750 (exon 19)	*TP53, ARID1A mutation* ***, MYC ampl, RICTOR ampl, TERT ampl*	osimertinib	2	BR: PR PFS: 6 months ***
Okamoto et al. [39] Araki et al. [40]	*EGFR* delE746-A750 (exon 19)	NA	gefitinib	1	BR: PR PFS: 5 months
Shiao et al. [37]	*EGFR* delE746-A750 (exon 19)	NA	gefitinib	4	BR: PD
Sun et al. [12]	*EGFR* exon 19 del	*RET* E616K (unk biological effect), multiple *TP53* mutations	gefitinib	NA	No response
Le et al. [41]	*EGFR* delL747_P753insS (exon 19)	NA	erlotinib	2	BR: PD
Petricevic et al. [42]	*EGFR* exon19 del	*TP53* and *PIK3CA* mutations	erlotinib	2	BR: PD
Sun et al. [46]	ALK rearrangement	*TP53* mutation, *RB1* truncating mutation	alectinib + irinotecan	2	BR: PR PFS: 6 months

* Cases of tumors with two components: adenocarcinoma and SCLC. ** ARID1A mutation was detected only in SCLC component; *** censored in August 2020, treatment ongoing. Abbreviations: Ampl: amplification; BR: best response; del: deletion; NA: not available; PD: progressive disease; PFS: progression-free survival; PR: partial response; unk: unknown.

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
