# Peer review of "Molecular and Genetic Advances in Small Cell Lung Cancer Landscape: From Homogeneity to Diversity"

_ijms, 2023, doi:10.3390/ijms25010224_

Round 1
Reviewer 1 Report
Comments and Suggestions for Authors
The review paper titled "Molecular and Genetic Advances in Small Cell Lung Cancer (SCLC) Landscape: From Homogeneity to Diversity" provides a comprehensive overview of the recent advancements in the understanding of SCLC. The review is a high-quality scientific document that significantly contributes to the field of lung cancer research. It effectively captures the evolving understanding of SCLC, highlights the importance of recognizing its molecular diversity, and opens up new avenues for research and treatment strategies. The paper is logical, informative, and well-articulated, making it a valuable resource for researchers and clinicians in this domain. However, there are several comments that need to be addressed before the manuscript can be considered for acceptance.
1.Although the review provides a comprehensive overview of molecular and genetic advancements in SCLC, its focus on basic science might not adequately address the practical, clinical implications of these findings. Clinicians and healthcare practitioners would benefit from deeper insights into how these molecular discoveries translate into tangible treatment strategies and ultimately improve patient care.
2.While the review mentions the classification into subtypes (SCLC-A, SCLC-N, SCLC-Y, and SCLC-P), a more detailed discussion on the characteristics, prevalence, and clinical implications of each subtype might enhance the utility of the paper. This would particularly benefit readers seeking in-depth insights into specific subtypes.
3.The paper highlights the potential of liquid biopsy in understanding resistance mechanisms to treatments. However, a more detailed exploration of known resistance mechanisms in SCLC, and how these findings could influence future therapeutic strategies, would strengthen the paper.
4.Provide a more detailed historical perspective on the study and treatment of SCLC. This could include the evolution of understanding from its initial discovery to the present, noting key milestones in research and treatment that have shaped current knowledge.
5.Include more comprehensive epidemiological data, such as incidence rates, mortality rates, demographic variations, and risk factors associated with SCLC. This data can help contextualize the disease's impact and relevance in a global health setting.
6.Offer a clearer differentiation between SCLC and other types of lung cancers, such as Non-Small Cell Lung Cancer (NSCLC). Highlighting the unique challenges, prognosis, and treatment strategies associated with SCLC compared to other lung cancers can provide a more robust understanding.
7.Discuss the evolution of standard treatments for SCLC over time, including initial approaches and how they have changed due to advancements in understanding the disease's molecular and genetic aspects.
8.Provide an overview of typical patient outcomes with current treatment modalities, discussing both successes and ongoing challenges. This sets the stage for understanding the need for new research and treatments.
Comments on the Quality of English LanguageNot required
Author Response
1.Although the review provides a comprehensive overview of molecular and genetic advancements in SCLC, its focus on basic science might not adequately address the practical, clinical implications of these findings. Clinicians and healthcare practitioners would benefit from deeper insights into how these molecular discoveries translate into tangible treatment strategies and ultimately improve patient care.
2.While the review mentions the classification into subtypes (SCLC-A, SCLC-N, SCLC-Y, and SCLC-P), a more detailed discussion on the characteristics, prevalence, and clinical implications of each subtype might enhance the utility of the paper. This would particularly benefit readers seeking in-depth insights into specific subtypes.
Joint response to points 1 & 2. The initial paragraph delves into the transcriptional and molecular subtypes of Small Cell Lung Cancer (SCLC), offering a rich array of basic science data. We have endeavored to enhance the readability of this section for physicians, while simultaneously imparting essential information, such as the most and least common subtypes. Furthermore, we have emphasized the possibilities for personalized treatment arising from the varying gene expression profiles across different subtypes. Our goal is to render our work less difficult to read and more aligned with a bench-to-bedside approach, effectively bridging the gap between discovery in translational science and clinical outcomes for patients
3.The paper highlights the potential of liquid biopsy in understanding resistance mechanisms to treatments. However, a more detailed exploration of known resistance mechanisms in SCLC, and how these findings could influence future therapeutic strategies, would strengthen the paper.
Thank you for the suggestion. In the liquid biopsy section, we concluded with a paragraph dedicated to CTCs-derived xenografts and the possibility of using these models for elucidating mechanisms of resistance in SCLC and studying new strategies (i.e. multiple blockages of different pathways) to overcome platinum resistance.
4.Provide a more detailed historical perspective on the study and treatment of SCLC. This could include the evolution of understanding from its initial discovery to the present, noting key milestones in research and treatment that have shaped current knowledge.
5.Include more comprehensive epidemiological data, such as incidence rates, mortality rates, demographic variations, and risk factors associated with SCLC. This data can help contextualize the disease's impact and relevance in a global health setting.
6.Offer a clearer differentiation between SCLC and other types of lung cancers, such as Non-Small Cell Lung Cancer (NSCLC). Highlighting the unique challenges, prognosis, and treatment strategies associated with SCLC compared to other lung cancers can provide a more robust understanding.
7.Discuss the evolution of standard treatments for SCLC over time, including initial approaches and how they have changed due to advancements in understanding the disease's molecular and genetic aspects.
8.Provide an overview of typical patient outcomes with current treatment modalities, discussing both successes and ongoing challenges. This sets the stage for understanding the need for new research and treatments.
Response to points 4-8. Dear reviewer, thank you for your suggestions. We re-organized the introduction, in order to address all these points and novel challenges in SCLC management.
“Introduction
Small Cell Lung Cancer (SCLC) accounts for 10-15% all lung cancer cases, with a prevalence of 1-5 cases per 10.000 people in Europe, and it is characterized by the strongest connection with smoking habit among all the lung cancer histologies(1).
Due to rapid growth and early metastatic spread, SCLC does not benefit from low dose computed tomography (CT) screening(2,3). Overall survival (OS) for the extended-disease (ED) is extremely poor (<10% at five years)(1). Unlike what has occurred with non-small cell lung cancer (NSCLC), the introduction of novel therapies into the treatment paradigm has been limited, and even when progress has been made, as with immunotherapy, the benefits achieved have generally been modest(1,4–6).
SCLC has traditionally been viewed as a homogeneous disease, shaping treatment strategies accordingly. The standard approach has involved platinum-based chemotherapy, resulting in rapid and profound responses but rarely achieving long-term durability(1). Recent translational research has begun to challenge this paradigm, fueling increasing interest in the molecular subtypes of SCLC and their potential implications for therapeutic strategies. Indeed, differential gene expression in the different molecular subtypes, and during disease course, might influence sensitivity and resistance to several therapeutic agents. The identification of predictors of response to immunotherapy is of extreme clinical importance, since the addition of an anti-Programmed-death ligand 1 (anti PD-L1) to the platinum-etoposide backbone represents the new standard of care for frontline treatment of ED-SCLC, regardless of PD-L1 or any other biomarker(4,5). The addition of atezolizumab and durvalumab to first-line chemotherapy treatment allows for a gain of two-three months in overall survival (OS), increasing from 10 to 12-13 months of survival. However, currently, there are no validated biomarkers that can identify those (few) patients who may achieve long-term benefit from frontline chemo-immunotherapy(4,5). Furthermore, novel targets, such as delta-like ligand 3 (DLL3), are of great interest for the potential development of new classes of drugs (antibody-drug conjugates, bispecific proteins) that specifically recognize them(7,8).
Finally, a small but significant percentage of SCLC cases occur in patients with no history of smoking, often with recognized oncogenic drivers (e.g., EGFR sensitizing mutations)(9–12). Due to the rarity of this occurrence, reports on disease course and treatment responses are mostly anecdotal and provide limited information
This review summarizes relevant updates involving the transcriptomic and genomic characterization of SCLC, both in terms of biological findings and clinical implications, with the objective of defining the evolving molecular landscape of this disease. Within the context of this review, we included a specific focus on potentially actionable oncogenic drivers and novel targets for drug development.”
Reviewer 2 Report
Comments and Suggestions for Authors
This review manuscript titled "Molecular and genetic advances in Small Cell Lung Cancer (SCLC) landscape: from homogeneity to diversity" explores the evolving understanding of SCLC as a disease, challenging its traditional classification as a homogeneous entity. It highlights the significance of recent molecular studies that reveal diverse genomic alterations beyond the well-known TP53 and RB1 mutations, categorizing SCLC into subtypes based on molecular characteristics. The article discusses the potential therapeutic implications of these molecular subtypes, emphasizing the need for personalized treatment strategies. Additionally, it addresses the challenges in targeting specific mutations, such as EGFR and ALK, in SCLC and suggests the emerging role of liquid biopsy as a valuable tool for understanding the molecular landscape and guiding treatment decisions.
Overall, the article is very informative and beneficial for readers. However, some improvements can be made to improve the clarity and readability of the text.
Comments:
- The paper the review could benefit from a more concise and structured presentation. The information is dense and might be challenging for some readers to follow due to the extensive use of technical terms and complex molecular details. Consider eliminating unnecessary technical details and focusing on the most critical information to improve overall readability. Another suggestion is providing a clearer roadmap for readers to follow the complex molecular details. (A summarized graph for this review would be better).
- Providing more context about the limitations and future directions would be valuable. Acknowledging potential biases, sample size issues, or other methodological concerns in the discussed research would contribute to a more well-rounded evaluation of the current state of SCLC research.
- This review could strengthen its discussion on the clinical implications of the molecular findings. Although it mentions the potential therapeutic applications of the identified molecular subtypes, more emphasis on how these findings may translate into improved patient outcomes or treatment strategies would enhance the practical relevance of the review. For example, a) the development of targeted agents or novel drug classes, b) the potential of liquid biopsy as a routine tool for baseline therapeutic decisions, c) the integration of immunotherapy into SCLC treatment regimens, d) identifying additional biomarkers.
Extensive editing of English language required.
Author Response
- The paper the review could benefit from a more concise and structured presentation. The information is dense and might be challenging for some readers to follow due to the extensive use of technical terms and complex molecular details. Consider eliminating unnecessary technical details and focusing on the most critical information to improve overall readability. Another suggestion is providing a clearer roadmap for readers to follow the complex molecular details. (A summarized graph for this review would be better). Thank you for the suggestion. We tried to simplify the most technical parts (especially the first paragraph), while maintaining the most relevant translational research findings. In order to improve readability, we stated at the beginning of the first paragraph that we would focus on the transcriptional factor-based categorization, since the heterogeneity of the disease mostly lies on their differential gene expression.
- Providing more context about the limitations and future directions would be valuable. Acknowledging potential biases, sample size issues, or other methodological concerns in the discussed research would contribute to a more well-rounded evaluation of the current state of SCLC research. We highlighted the limited sample size of most of liquid biopsy studies in the discussion section, with the consequential limitation in drawing conclusion from them and translating results into clinical practice. We also highlighted, all over the manuscript, the retrospective nature of the reports regarding the occurrence of SCLC, with or without oncogenic drivers, in the absence of smoking history.
- This review could strengthen its discussion on the clinical implications of the molecular findings. Although it mentions the potential therapeutic applications of the identified molecular subtypes, more emphasis on how these findings may translate into improved patient outcomes or treatment strategies would enhance the practical relevance of the review. For example, a) the development of targeted agents or novel drug classes, b) the potential of liquid biopsy as a routine tool for baseline therapeutic decisions, c) the integration of immunotherapy into SCLC treatment regimens, d) identifying additional biomarkers. Thank you for the suggestion. We re-organized the discussion in order to give more space and emphasis to the possibility of personalizing treatment through the SCLC-A/N/P/I categorization and liquid biopsy. We discussed differential sensitivity to chemotherapy and other agents (PARP-inhibitors), potential novel targets (DLL3), predictive biomarkers for durable response to immunotherapy (MLH-1 expression and inflamed microenvironment).
Reviewer 3 Report
Comments and Suggestions for Authors
Fairly well written and informative narrative review on the molecular features of SCLC.
The following minor adjustments are needed to improve this manuscript:
Line 65, “TP53 being fundamental to arrest or induce …” should be “TP53 being fundamental to arrest cell proliferation or induce …”
Line 76, to be consistent with the other mentioned transcriptional regulators, the abbreviation of the transcription factor “ASCL1” should be defined as “Achaete-Scute Family BHLH Transcription Factor 1” first. By the same token, “neuronal differentiation 1” should be written before “NeuroD1”.
Line 102, “POU2F3 positive and negative (10)” should be “POU2F3 positive and negative SCLC (10)”.
Line 104, “was capable to distinguish” should be “was capable of distinguishing” or “was able to distinguish”.
Line 118, “characterized by a higher immune infiltration” would be more appropriate as “characterized by a higher infiltration of immune cells”
Line 126, “Some attempts have been taken”: Some attempts have been made.
Line 130: correct the typo “an higher”.
Line 134, “through Foundation Medicine Inc. panel,”: the panel is an NGS panel, and the corresponding reference is # 18. These two things should be written in the text here for clarity. Right now, the citation of reference 18 is placed at the end of the description of Sivakumar et al.’s results on line 160.
Reference 17 is the same as reference 6. The order of cited references and reference list should be changed accordingly.
Line 238-9, “histological transformation as a mechanism of resistnce to tyrosine kinase inhibitors (TKIs)(25–29)”: Reference 29 deals with transformation of an ALK+ LUAD to SqCC, not SCLC, thus it is not appropriate here. More suited references should be cited, for ex. the extensive review on transformation of ALK+ NSCLC to SCLC in doi: 10.21037/tlcr-22-867
Line 156-7, “through the E7 156 oncoprotein”: it should be explained better as HPV produces two oncoproteins. E6 binds and inhibits p53, E7 binds and inhibits Rb.
Line 175-6: it should be “three PIK3CA mutations” and “six PIK3CA amplifications”
Line 223: correct “and and those harboring”
Line 239: correct “resistnce”: line 240: “de-novo” should be “de novo”
Line 302: adjust “mo(41,42)lecular”
Line 305-7 referring to a case of ALK-mutated small cell carcinoma of the prostate: it this relevant for this review on SCLC? Prostate cancer and lung cancer are totally different diseases, also molecularly.
Line 396: adjust “forward(61)d”.
Reference 50: The first author's name is missing in the reference list.
Comments on the Quality of English LanguageThere are quite a few typos and some phrasing errors. Related adjustments are suggested in Comments and Suggestions for Authors.
Author Response
Dear reviewer,
We revised the paper and corrected all the typos/mispelled words.
Moreover:
1) "Line 134, “through Foundation Medicine Inc. panel,”: the panel is an NGS panel, and the corresponding reference is # 18. These two things should be written in the text here for clarity. Right now, the citation of reference 18 is placed at the end of the description of Sivakumar et al.’s results on line 160." It has been corrected
2) "Line 238-9, “histological transformation as a mechanism of resistnce to tyrosine kinase inhibitors (TKIs)(25–29)”: Reference 29 deals with transformation of an ALK+ LUAD to SqCC, not SCLC, thus it is not appropriate here. More suited references should be cited, for ex. the extensive review on transformation of ALK+ NSCLC to SCLC in doi: 10.21037/tlcr-22-867" Thank you for the suggestion. We substituted that reference with a more appropriate one.
3) "Line 305-7 referring to a case of ALK-mutated small cell carcinoma of the prostate: it this relevant for this review on SCLC? Prostate cancer and lung cancer are totally different diseases, also molecularly. " We erased this paragraph.
4) "Line 156-7, “through the E7 156 oncoprotein”: it should be explained better as HPV produces two oncoproteins. E6 binds and inhibits p53, E7 binds and inhibits Rb." It has been explained: "The authors explored the potential role of human papilloma virus (HPV) infection in the functional inactivation of p53 and Rb, specifically through the E7 oncoprotein, which inactivates RB-family p107 and p130. "